# Effect of Chilling Temperature on Survival and Post-Diapause Development of Korean Population of *Lymantria dispar asiatica* (Lepidoptera: Erebidae) Eggs

**Min-Jung Kim** [1], **Keonhee E. Kim** [1], **Cha Young Lee** [1,2], **Yonghwan Park** [1], **Jong-Kook Jung** [1,2,*] **and Youngwoo Nam** [1,*]

1   Forest Entomology and Pathology Division, National Institute of Forest Science, Seoul 02455, Republic of Korea
2   Department of Forest Environment Protection, Kangwon National University, Chuncheon 24341, Republic of Korea
*   Correspondence: jkjung@kangwon.ac.kr (J.-K.J.); orangmania99@korea.kr (Y.N.)

**Abstract:** One of the subspecies of the Eurasian spongy moth, *Lymantria dispar asiatica*, is a destructive forest pest in native regions and also an important quarantine pest in non-native regions. Its polyphagous nature, together with occasional outbreaks, may seriously threaten ecosystems and result in costly management programs. In this study, we examined the effect of chilling temperatures (−12, −6, 0, 6, and 12 °C) during the diapause phase on the survival and post-diapause development of *L. d. asiatica* eggs, collected before winter, in order to characterize their thermal response. The eggs were exposed to treatment temperatures for 100 days, followed by 25 °C incubation to determine their survival and development time. The eggs hatched in all the treatments, indicating that all the examined conditions could partly or sufficiently satisfy the thermal requirement for eggs to enter post-diapause development. However, exposure to chilling temperatures significantly affected both the survival and development times of overwintering eggs in a given temperature range. The survival rates declined at −12 °C, and the development rates accelerated as the chilling temperature increased. This information could offer clues for the assessment of the outbreak potential in native regions and the possibility of range expansion in non-native regions through the consideration of winter conditions that favor *L. d. asiatica* egg hatching and their subsequent development.

**Keywords:** spongy moth; gypsy moth; overwintering ecology; insect pest outbreak; egg diapause





## 1. Introduction

One of the subspecies of the Eurasian spongy moth, *Lymantria dispar asiatica* Vnukovskij, is native to temperate regions, including China, the Russian Far East, and Korea [1–3]. It is a destructive forest pest, due to its polyphagous characteristics, with occasional mass outbreaks. Its larvae feed on more than 300 deciduous and several coniferous tree species, so damage could occur in virtually all types of forests [4–6]. Massive outbreaks of *L. d. asiatica* has occasionally occurred in Korea [7,8]; in particular, severe defoliation caused by voracious larvae was observed in the whole territory, with the exception of the southwestern regions, in 2020 [8–10]. This defoliation can affect plant vitality and the forest ecosystem in large infestation areas [11,12]. Dulamsuren et al. [13] reported that trees defoliated by *L. d. asiatica* experience stunted growth for several years. Due to the economic and ecological impact of European subspecies *Lymantria dispar* L. in the invaded area, such as in North America, *L. d. asiatica* is also considered an important quarantine pest in non-native regions [8,14–16]. The flight ability of females and the broad host range of Asian strains may pose a serious threat to the ecosystem of such areas and result in costly management programs [14,15]. To prevent the introduction and local establishment of *L. d. asiatica* in the port area, North American countries have demanded a certificate of inspection of freedom for vessels passing through regions where the Asian strains are present [8].

The outbreak of insect pests is the result of complex interactions between factors such as weather, hosts, predators, parasites, disease, and heritable traits [17]. Among the various factors, temperature is one of the key aspects determining an insect's performance, including survival, emergence as a developmental stage, and host availability. In particular, the life history of *L. d. asiatica* indicates that the temperature at which the egg stage occurs is critical for its performance. *Lymantria dispar asiatica* is a univoltine species and spends most of its life in the egg stage [5,18,19]. Some eggs oviposited early in the season could hatch in the fall [20], but the egg stage is generally maintained for around nine months, from the end of July to early April, in temperate regions, including Korea [8,20]. Embryos enter the pre-diapause development phase in response to the higher temperature in the summer, and then diapause to endure low temperatures in the winter [21,22]. Many studies have suggested that the temperature during the diapause phase plays a pivotal role in the mortality and developmental rate of *L. d. asiatica* eggs and consequently acts as an ecological barrier delineating its geographical distribution [15,19,22]. Wei et al. [19] suggested that the inability to hatch without chill in the Chinese population could explain the distribution of *L. d. asiatica* in China. Proper exposure to low temperatures of a certain duration increases successful hatching and decreases the embryo development time. Thus, the winter chilling temperature could be a key factor in determining the initial population size and occurrence time of larvae for the year in terms of an outbreak [6,23,24]. The response of eggs relative to the chilling temperature would provide foreknowledge for the determination of the potential geographic range of *L. d. asiatica* [11,25,26].

The thermal responses of *L. d. asiatica* eggs in the diapause phase have been studied under certain temperatures and exposure periods [15,19]. Keena [15] reported that chilling periods of longer than 60 days at 5 °C were required for the successful hatching of a Russian population, maintained in the laboratory, when the eggs were subsequently incubated at 25 °C after chilling treatment. Similar results for Chinese populations collected from fields in February were reported, indicating that a low temperature (0 or 5 °C) with a certain duration that was not overly long could facilitate the post-diapause development of embryonated eggs [19]. These studies support the importance of the chilling requirement for successful egg development and highlight that a constant temperature greater than 15 °C causes delayed or failed hatching, and most eggs (>99%) cannot hatch at 25 °C. Ananke and Kolosov [26] evaluated various chilling temperature regimes, ranging between −23 and −29.9 °C, focusing on the estimation of the lowest lethal temperature for the Central Asian population. In the field, however, eggs will experience varied chilling conditions, not only extremes or preferable temperatures. Moreover, physiological responses have been shown to differ between Asian and European strains and even between regional populations within a single strain [15,22,27]. These factors make it difficult to understand the thermal responses and population dynamics in order to estimate the outbreak potential of *L. d. asiatica*, depending on the region and year.

In this study, we investigated the thermal responses of field-collected *L. d. asiatica* eggs at different chilling temperatures covering the actual range of winter conditions in Korea. The results from this study can contribute to a more reliable assessment of the initial population size and its occurrence time in the field.

## 2. Materials and Methods

### 2.1. Collection of Overwintering Eggs

Egg masses of *L. d. asiatica* were collected from eight regions in Korea during November 2020 (Table 1; Figure 1). We visited the locations where adults of *L. d. asiatica* had been widely observed in the summer of 2020. The egg masses were carefully scraped from the tree trunks at the sampling site. To obtain viable eggs for our experiments, egg masses with emergence holes (e.g., oviposited in the past year or with emerged egg parasitoids) were not chosen during the sampling. We assumed that the eggs' pre-diapause development had terminated in all sampling periods because the summer temperature of Korea is sufficient to fulfill the reported heat requirement, such as 27 days at 25 °C or 48 days at 15 °C [24]

(Table 1). Egg populations collected from Gunpo (GP), Chuncheon (CC), Danyang (DY), and Chungju (CJ) were exposed to experimental temperatures on the sampling dates. Egg samples from Hongcheon (HC), Yeongyang (YY), Cheongsong (CS), and Andong (AD) started treatment at experimental temperatures within three days of the sampling date. These eggs were maintained at 0 °C before experimental treatments.

**Table 1.** Sampling information for the overwintering eggs of *Lymantria dispar asiatica*.

| Site Name | Latitude and Longitude (°, min, s) | Collection Date | Mean Temperature (°C) | | | | No. Days (≤12 °C) [2] |
|---|---|---|---|---|---|---|---|
| | | | August | September | October | November [1] | |
| Gunpo (GP) | 37°19′41.9″ N, 126°54′38.9″ E | 4 November | 26.3 | 21.1 | 14.3 | 9.8 | 7 |
| Hongcheon (HC) | 37°41′54.2″ N, 127°57′36.4″ E | 17 November | 24.8 | 18.1 | 10.3 | 5.9 | 36 |
| Chuncheon (CC) | 37°53′37.3″ N, 127°41′31.2″ E | 18 November | 26.1 | 19.7 | 11.9 | 7.8 | 35 |
| Danyang (DY) | 36°58′24.2″ N, 128°25′06.2″ E | 18 November | 25.8 | 19.2 | 12.2 | 8.3 | 33 |
| Chungju (CJ) | 36°57′52.7″ N, 127°57′31.7″ E | 18 November | 27.7 | 21.5 | 14.5 | 10.4 | 18 |
| Yeongyang (YY) | 36°39′28.8″ N, 129°07′13.1″ E | 28 November | 25.6 | 18.7 | 11.9 | 7.1 | 44 |
| Cheongsong (CS) | 36°25′58.1″ N, 129°03′48.6″ E | 28 November | 25.7 | 18.5 | 11.6 | 7.1 | 43 |
| Andong (AD) | 36°32′57.5″ N, 128°43′34.3″ E | 28 November | 26.6 | 19.6 | 12.7 | 7.9 | 39 |

[1] Mean temperature from November 1 to sampling date of each site; [2] Number of days with daily mean temperature below 12 °C.

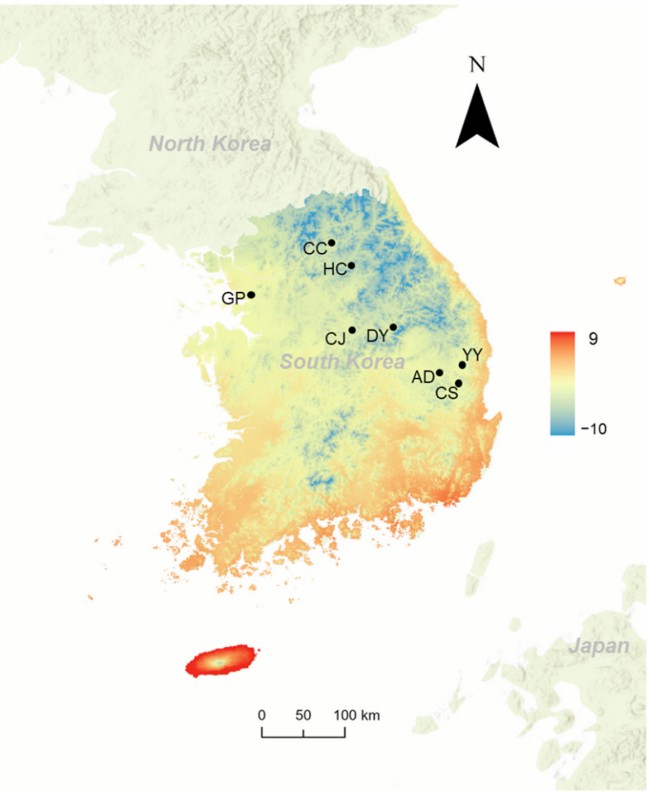

**Figure 1.** Sampled location for *L. d. asiatica* eggs in this study on the distribution map of mean temperature of coldest quarter (https://www.worldclim.org/, accessed on 31 October 2022). Abbreviations of locations are given in Table 1.

## 2.2. Experimental Design

To examine the effect of chilling temperatures on the survival and post-diapause development of *L. d. asiatica*, the collected eggs were exposed to five different low temperatures (−12, −6, 0, 6, and 12 °C) for 100 days, with approximately 50%–60% relative humidity (RH), in growth chambers (for 0, 6, and 12 °C; DS-8CL, Dasol Scientific; Hwaseong, Korea) and freezers (for −12 and −6 °C; C053AF, LG Electronics; Seoul, Korea) (Figure 2). The eggs were treated in the form of egg masses, rather than detached individuals, in order to minimize exposure to room temperature as a result of the handling process. The experimental temperatures were set to cover the actual range of winter temperatures in Korea (Korea Meteorological Administration; KMA). The duration of low temperature exposure in our experiments was determined by considering the length of the winter period, from December to February, in Korea and the duration of the chilling requirement under which stable survival was expected [14]. The egg masses collected at each site were randomly transferred to an insect breeding dish (Cat. No. 310202, SPL; Pocheon, Korea) with a filter paper (90 mm in diameter) (Cat. No. WF6-0900, Hwan-gyeong Tec; Seoul, Korea). Then, the dishes were allocated into chambers at the five temperatures. At least 15 egg masses were tested for each population and experimental temperature.

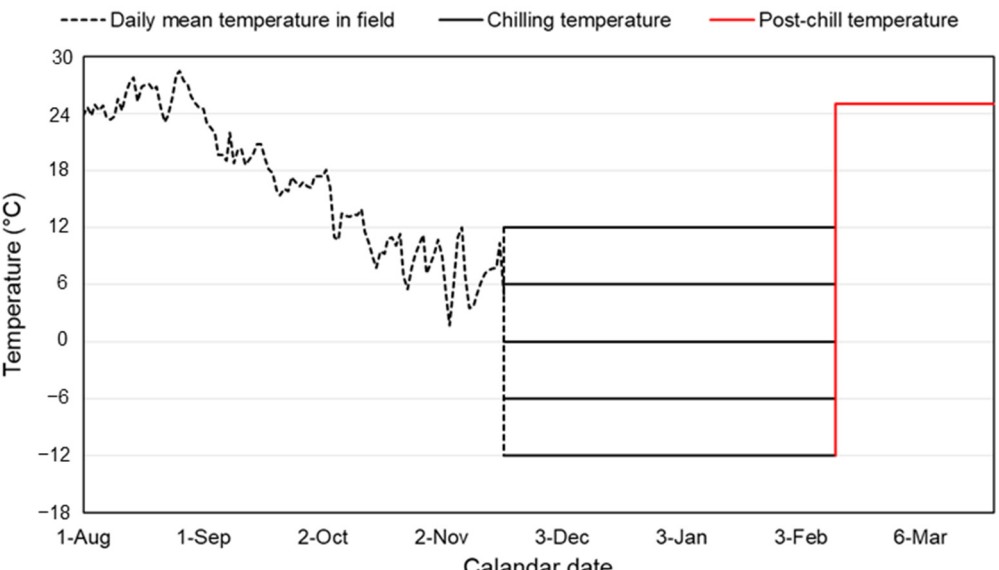

**Figure 2.** Schematic diagram of temporal pattern of temperature experienced by eggs of HC population in this study. Daily mean temperatures before the date of egg collection were obtained from KMA website (http://web.kma.go.kr/, accessed on 31 October 2022). The eggs were exposed to one of the five chilling temperatures (−12, −6, 0, 6, and 12 °C) for 100 days, followed by post-chill temperature of 25 °C.

Following chilling treatment for 100 days, the insect breeding dishes with egg samples were exposed at 25 °C with 70%–80% relative humidity (RH) and a 12:12 h (L:D) cycle to induce the post-diapause development of eggs [14,19] (Figure 2). The post-chill treatment was conducted in a growth chamber (DS-8CL, Dasol Scientific; Hwaseong, Korea), and floral foam (Oasis, Smithers-oasis Korea; Cheonan, Korea) sufficiently soaked in tap water was placed inside to maintain the RH. In the 12 °C chilling treatment, the egg hatching had already progressed; thus, egg masses including unhatched individuals were incubated at 25 °C after the emerged larvae were counted and removed from the dishes. Egg development time (days) and the number of hatches were recorded, every day, for each treatment. The daily observations were conducted until there was no hatching for two weeks from the last hatching for each treatment temperature and population. The egg samples that had been checked for hatching were dehaired, and the remaining eggs (i.e., unhatched individuals) were counted to determine the survival rate. During the experiments with

incubation, parasitic wasps emerged, with the exception of the CS population. The proportion of emerged egg parasitoid wasps to the total (i.e., wasp + larva + remaining egg) varied according to the population, ranging between 0 and 2.4%. However, the number of wasps was not included in the total valid eggs in determining the survival rate of *L. d. asiatica* eggs because the aim of this study was to evaluate the performance of our target species, and not the success rate of parasitism.

### 2.3. Data Analysis

The fate of the eggs in the experiments was recorded as frequency data of binomial classes (i.e., hatching or unhatched), including the number of hatches at 12 °C before 25 °C incubation. Then, the effect of the chilling temperatures on the survival of the *L. d. asiatica* eggs was assessed using the generalized linear mixed effect model (GLMM) with binomial distribution. Temperature treatment combination was included as a fixed effect, and sampling site was considered as a random effect in the model. GLMM analysis was performed using the function 'glmer' in the *lme4* package in R 4.1.1 [28,29]. To evaluate the effect of the chilling temperatures on the post-diapause development of *L. d. asiatica*, linear mixed effect model (LMM) analysis, incorporating random effects, was applied. We considered treatment temperature and sampling site as fixed and random effects, respectively. However, the development time in the 12 °C treatment was excluded from the analysis because the time in days could not be determined by checking the hatched eggs at the end of treatment and before the incubation at 25 °C. An LMM model was constructed using the 'lmer' function in the *lme4* package [28]. The significances of the fixed variable (i.e., temperature) in the GLMM and LMM models were determined by a Type II Wald chi-square test using the 'Anova' function in the *car* package. Pairwise comparisons were performed using Tukey contrast with the 'glht' and 'mcp' functions in the *multicomp* package in R 4.1.1.

The effect of the chilling temperatures within a sampling site with different collection times was also evaluated. The survival of each site was assessed using the chi-square test, followed by a post hoc comparison with Bonferroni correction. The analysis was conducted using the functions in the *stata* and *rcompanion* packages in R 4.1.1. Mean development times (day) of each site were separated based on the post hoc comparison using Tukey's honest significant difference test. Variabilities in the hatchability of the *L. d. asiatica* eggs between sampling sites were compared using the coefficient variations (CV, standard deviation/mean) of survival rate and median development time of the eight sites.

The variation in development time, according to the treatment, was described. As the development time at 12 °C of chilling is inevitably censored, due to the unclear first date of hatching, we compared the cumulative proportion of hatching to capture the effect of the temperature treatment in the development time distribution. After the experiment, the cumulative proportions for each treatment were pooled and fitted against the days using the two-parameter Weibull cumulative function:

$$y = 1 - Exp[(-x/a)^b] \tag{1}$$

where *y* is the cumulative proportion of hatches at a development time *x* in days, and *a* and *b* are the scale and shape parameters of the Weibull function, respectively. The scale parameter *a* represents the time of the 63.2 percentile of hatching in the model. The model parameters were estimated using the 'nls' function in the *stata* package in R 4.1.1 [29].

### 3. Results

*Lymantria dispar asiatica* eggs experiencing temperatures from −12 to 12 °C for 100 days could develop to the first instar during chilling treatment or at 25 °C after the treatment. However, the survival of eggs relative to the temperature treatments was significantly different (Wald $\chi^2$ = 17,691; df = 4; $p < 0.0001$) (Figure 3A). The survival rate was significantly higher at −6, 0, and 6 °C than at the extreme temperature of −12 °C. In comparison within the sapling site, there was also a significant difference in the survival rate according to

temperature treatment in all populations (GP: $\chi^2$ = 8711.9; df = 4; $p < 0.0001$; HC: $\chi^2$ = 2377.8; df = 4; $p < 0.0001$; CC: $\chi^2$ = 892.3; df = 4; $p < 0.0001$; DY: $\chi^2$ = 4087.7; df = 4; $p < 0.0001$; CJ: $\chi^2$ = 5161.4; df = 4; $p < 0.0001$; YY: $\chi^2$ = 1156.9; df = 4; $p < 0.0001$; CS: $\chi^2$ = 1558.3; df = 4; $p < 0.0001$; AD: $\chi^2$ = 3651.5; df = 4; $p < 0.0001$) (Table 2). The highest survival was generally observed at 0 °C or 6 °C, with the exception of the CC and DY populations, which showed the highest value at −6 °C (Table 2).

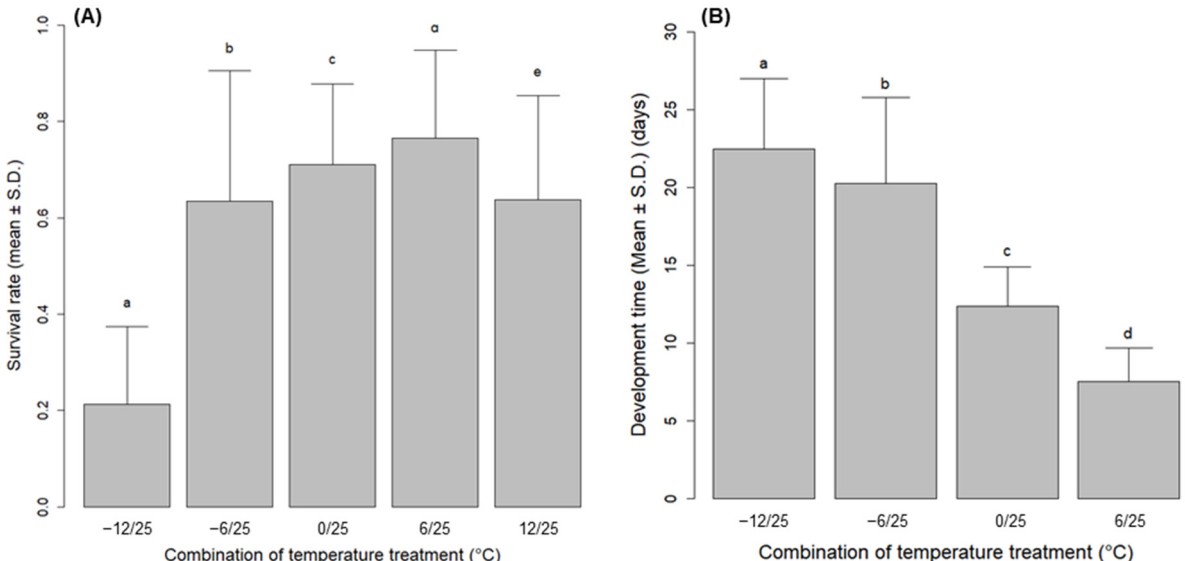

**Figure 3.** (**A**) survival rate and (**B**) development time of *Lymantria dispar asiatica* eggs of eight experimental populations for each treatment. Different letters above the bars indicate significant differences according to mixed effect models ($p < 0.001$).

**Table 2.** Summary of survival rate and development time of *Lymantria dispar asiatica* eggs depending on exposure to chilling temperatures for 100 days, followed by 25 °C incubation.

| Site | Chilling Treatment (°C) | Sample Size [1] | Survival Rate (%) | Hatches before Incubation at 25 °C (% Proportion) | Development Time at 25 °C after Chilling (Mean ± S.D) (Days) |
|---|---|---|---|---|---|
| GP | −12 | 5087 | 20.7 [a,2] | 0 | 24.4 ± 4.47 [a,3] |
| | −6 | 4905 | 43.6 [b] | 0 | 25.4 ± 6.13 [b] |
| | 0 | 4768 | 83.5 [c] | 0 | 12.5 ± 2.36 [ac] |
| | 6 | 5104 | 92.6 [d] | 0 | 8.3 ± 2.48 [d] |
| | 12 | 12,993 | 76.6 [e] | 23.2 | NA [4] |
| HC | −12 | 2493 | 20.6 [a] | 0 | 21.6 ± 4.05 [a] |
| | −6 | 853 | 85.5 [b] | 0 | 20.6 ± 4.13 [b] |
| | 0 | 719 | 91.9 [c] | 0 | 11.9 ± 1.49 [c] |
| | 6 | 803 | 92.9 [c] | 0 | 6.3 ± 1.39 [d] |
| | 12 | 292 | 61.0 [d] | 57.9 | NA |
| CC | −12 | 735 | 19.6 [a] | 0 | 22.7 ± 4.55 [a] |
| | −6 | 725 | 83.2 [b] | 0 | 17.7 ± 2.87 [b] |
| | 0 | 517 | 54.4 [c] | 0 | 14.8 ± 3.08 [c] |
| | 6 | 712 | 78.5 [b] | 0 | 7.4 ± 1.28 [d] |
| | 12 | 500 | 82.8 [b] | 69.0 | NA |
| DY | −12 | 3996 | 57.5 [a] | 0 | 21.2 ± 4.48 [a] |
| | −6 | 4794 | 96.1 [b] | 0 | 16.5 ± 3.74 [b] |
| | 0 | 4293 | 94.9 [b] | 0 | 10.4 ± 1.56 [c] |
| | 6 | 4138 | 90.7 [c] | 0 | 6.5 ± 1.64 [d] |
| | 12 | 17,241 | 90.2 [c] | 75.9 | NA |

**Table 2.** *Cont.*

| Site | Chilling Treatment (°C) | Sample Size [1] | Survival Rate (%) | Hatches before Incubation at 25 °C (% Proportion) | Development Time at 25 °C after Chilling (Mean ± S.D) (Days) |
|------|------|------|------|------|------|
| CJ | −12 | 3061 | 4.3 [a] | 0 | 24.7 ± 5.07 [a] |
|    | −6  | 3163 | 45.4 [b] | 0 | 22.1 ± 6.04 [b] |
|    | 0   | 3518 | 68.7 [c] | 0 | 12.8 ± 2.11 [c] |
|    | 6   | 3522 | 83.0 [d] | 0 | 8.7 ± 2.02 [d] |
|    | 12  | 4135 | 71.6 [c] | 26.3 | NA |
| YY | −12 | 2800 | 12.8 [a] | 0 | 23.3 ± 4.38 [a] |
|    | −6  | 2279 | 26.4 [b] | 0 | 24.5 ± 5.79 [b] |
|    | 0   | 3042 | 51.9 [c] | 0 | 13.7 ± 2.54 [c] |
|    | 6   | 2888 | 42.9 [d] | 0 | 6.1 ± 1.45 [d] |
|    | 12  | 7884 | 22.6 [e] | 14.0 | NA |
| CS | −12 | 2486 | 10.3 [a] | 0 | 23.4 ± 4.39 [a] |
|    | −6  | 2455 | 40.2 [b] | 0 | 22.4 ± 4.02 [b] |
|    | 0   | 1814 | 60.5 [c] | 0 | 13.5 ± 2.31 [c] |
|    | 6   | 2961 | 55.9 [d] | 0 | 7.4 ± 1.93 [d] |
|    | 12  | 2973 | 48.8 [e] | 43.0 | NA |
| AD | −12 | 3882 | 24.8 [a] | 0 | 22.9 ± 3.75 [a] |
|    | −6  | 3970 | 87.4 [b] | 0 | 20.4 ± 3.67 [b] |
|    | 0   | 3686 | 62.6 [c] | 0 | 13.6 ± 2.7 [c] |
|    | 6   | 3331 | 76.0 [d] | 0 | 7.1 ± 1.91 [d] |
|    | 12  | 7331 | 57.3 [e] | 49.9 | NA |

[1] Numbers represent the total valid eggs, which is the sum of total hatched larvae and unhatched eggs. [2] Numbers followed by the same letter within a column in a site are not significantly different according to the chi-square test ($p \geq 0.05$). [3] Means followed by the same letter within a column in a site are not significantly different according to Tukey's honest significant difference test ($p \geq 0.05$). [4] The development time in the 12 °C treatment was excluded from the analysis because the time in days could not be determined by checking the hatched eggs at the end of treatment and before incubation at 25 °C.

At 12 °C, a substantial number of eggs, ranging between 14.0 and 75.9%, of total successful hatches in the treatment hatched before being moved to 25 °C (Table 2). In other treatments, the development time of the eggs significantly differed depending on the treated chilling temperature (Wald $\chi^2$ = 160,660; df = 3; $p < 0.0001$) (Figure 3B). This difference was consistent across the within-group comparisons in all populations (GP: $F$ = 14,307; df = 3, 11,892; $p < 0.0001$; HC: $F$ = 4024; df = 3, 2645; $p < 0.0001$; CC: $F$ = 1984; df = 3, 1583; $p < 0.0001$; DY: $F$ = 14,854; df = 3, 14,723; $p < 0.0001$; CJ: $F$ = 5585; df = 3, 6904; $p < 0.0001$; YY: $F$ = 5547; df = 3, 3773; $p < 0.0001$; CS: $F$ = 6620; df = 3, 3994; $p < 0.0001$; AD: $F$ = 11,550; df = 3, 9262; $p < 0.0001$) (Table 2). There was a pattern of decreasing development time as the chilling temperature increased in the observation range (Figure 3B and Table 2). The variations in survival rate and median development time of *L. d. asiatica* eggs between sampling sites were lowest at the chilling temperature of −12 °C (Figure 4).

The development completion models clearly show the effect of chilling temperatures on the distribution of hatched eggs (Figure 5). As the temperature increased in the range of −12 to 12 °C, the hatching of *L. d. asiatica* accelerated overall, as seen in the parameter *a* of the model, which represents the time of the 63.2 percentile of hatching (Table 3). The development was essentially the fastest at 12 °C, but it was considerably delayed in some individuals (Figure 5). For example, the last hatch in the DD population was 153 days after the start of the experiment.

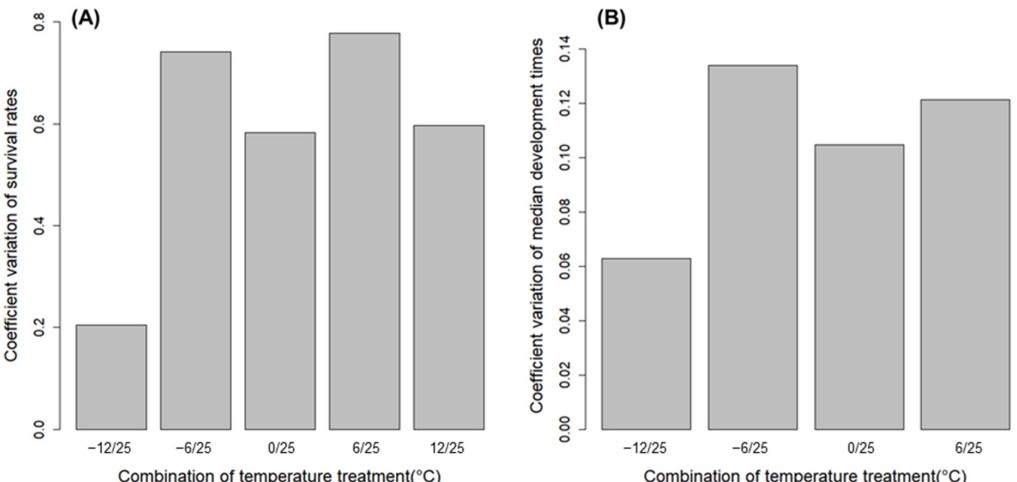

**Figure 4.** Coefficient variation (CV) of (**A**) survival rate and (**B**) median development time of *Lymantria dispar asiatica* eggs of sampled sites (*n* = 8) for each treatment.

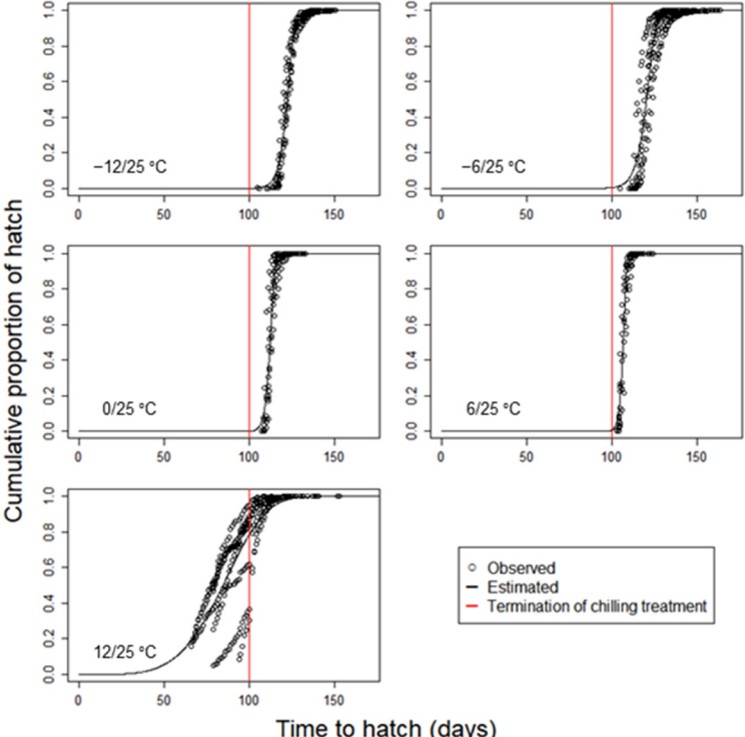

**Figure 5.** Cumulative distribution models for the development completion of *Lymantria dispar asiatica* eggs depending on different chilling treatments.

**Table 3.** Estimated parameter values of development distribution model for *Lymantria dispar* asiatica eggs depending on chilling temperature.

| Chilling Temperature (°C) | Parameter | | $r^2$ |
|---|---|---|---|
| | *a* | *b* | |
| −12 | 123.586 ± 0.1054 | 36.270 ± 1.4439 | 0.96 |
| −6 | 121.988 ± 0.2157 | 27.757 ± 1.8266 | 0.85 |
| 0 | 113.043 ± 0.1282 | 54.517 ± 4.4040 | 0.89 |
| 6 | 107.273 ± 0.1025 | 68.075 ± 5.6031 | 0.91 |
| 12 | 91.257 ± 0.5851 | 4.745 ± 0.2610 | 0.69 |

## 4. Discussion

In this study, we examined the effect of five temperatures, ranging between $-12$ and $12\,°C$, that *L. d. asiatica* eggs are expected to experience in the field during the winter period. As the overwintering eggs were collected from different sites with different dates, the eggs may have received different amounts of chill when the eggs were collected. The diapause state would thus not have been synchronized between the sampled populations; it might cause different responses of eggs according to the sampling site. For example, the mean development times at $6\,°C$ were correlated with the number of days, with daily mean temperatures below $12\,°C$ of the sampling sites (Pearson' $r = -0.76$; $p < 0.05$). Thus, the eggs experiencing more chill in the field conditions could more rapidly develop at $25\,°C$ after receiving an additional chill at $6\,°C$. However, there was no linear relationship between the development times at other temperatures and survival rates in all temperature treatments ($p > 0.05$). The variation was the lowest at $-12\,°C$, which may be because the chill treatment synchronized the state of eggs for development and survival by forcing deeper diapause [30]. On the other hand, the variation in the survival and development rates between sampling sites was relatively high at other temperatures, which may have originated from several factors besides the effect of the sampling dates. Firstly, the parasitism rate could affect the estimated survival rate of eggs, because it was uncertain whether the unhatched eggs in the experiments had been parasitized, although the proportion of emerged parasitic wasps was relatively low. In addition, environmental adaptation and genetic differences in the regional populations in Korea may be a factor causing different phenotypic responses, such as in thermal requirements and the proportion of non-diapause eggs in the population [31,32]. However, despite these fundamental variations originating from the field sampling, our study successfully captured the general thermal responses of *L. d. asiatica* eggs in the Korean population. Both the survival rate and the development time were significantly affected by the chilling temperature during the diapause phase. The survival rates declined at $-12\,°C$, and the development rates accelerated as the chilling temperature increased in all of the sampled populations.

In our experiments, many eggs collected in November were unexpectedly able to hatch at $12\,°C$ before 100 days and without incubation at $25\,°C$ (Table 2). Thus, a substantial proportion of eggs had already completed the diapause stage and began post-diapause development within 100 days, indicating that a temperature of $12\,°C$ could provide both chilling and heating units in succession. In addition, the results indicate that the lower development threshold temperature of overwintered eggs would posit between 6 and $12\,°C$. Thus, overwintered eggs would be able to begin development in the following March, and hatch in April, in Korea, when considering the estimated development threshold range and actual occurrence of first instar larvae. The period of 100 days at $12\,°C$ was not sufficient to hatch the entire population of overwintering eggs because some individuals required additional heat to hatch after 100 days. On the other hand, the overwintered eggs collected in mid-February could complete hatching within 100 days under relatively lower temperatures of $8\,°C$ (unpublished observation). Similarly, both Chinese populations collected in late January and early February terminated hatching within only 70 days at $10\,°C$, while fresh eggs that had not experienced low temperatures could not hatch within 200 days, even at higher temperatures of 20 and $25\,°C$ [19]. Keena [14] also reported that a Russian population maintained in the laboratory required approximately 200 days for the completion of hatching at 15 and $20\,°C$ after the termination of pre-diapause for 32 days at $25\,°C$. These phenomena indicate that the winter chilling temperature could facilitate the post-diapause development of *L. d. asiatica* by readying the eggs to develop in response to the preferable temperature in the spring season. In our experiments, the eggs could hatch under all chilling treatments, indicating that all examined conditions could partly or sufficiently satisfy the thermal requirement for eggs to enter post-diapause development. However, the temperatures, with the exception of $12\,°C$, would probably not be sufficiently high for the completion of egg development after the termination of diapause within 100 days as no hatching was observed before the post-chill treatment.

There was a considerable difference in the survival rate depending on the chilling temperature in our experiments. The survival rate slightly declined at an extreme of 12 °C, and this result partially matches that of a previous study reporting that the hatching rate of a Russian strain decreased with an increasing chilling temperature ranging between 5 and 15 °C in the diapause phase [14]. Although there were huge variations according to the sampled area, a decline in survival was also observed at −12 °C within one population. The excessively early completion of diapause might be responsible for the reduced hatching rate at −12 °C. This would cause depleted resources for post-diapause development during the holding periods [14]. On the other hand, the chilling temperatures of 0 and 6 °C showed a relatively stable performance in terms of the survival rate among the examined temperatures. This result suggests that the winter temperature range of Korea is currently suitable, overall, for the successful termination of diapause in *L. d. asiatica*. In addition, the result implies that eggs in the diapause phase are resistant to a broad temperature range for the whole winter period in temperate regions.

One of the new findings of our study is that *L. d. asiatica* eggs develop faster as the chilling temperature is increased, within a given exposure period, after pre-diapause phase. As there is no clear border between the diapause and post-diapause phases, individuals that experienced temperatures of 0 and 6 °C were mostly presumed to have terminated diapause within 100 days under chilling treatments and had initiated development for hatching. This is supported by the fact that many eggs treated at 12 °C could hatch within 100 days, and there was a relatively small variation within sampling sites in the development time at 0 and 6 °C, with the exception of the CC population (see Table 2). This is also in line with the fact that the longer the exposure time at the low temperature of 5 °C, the faster the development time in post-chill incubation [14,33]. At −6 and −12 °C, the treated eggs were also expected to have been provided with sufficient chilling within the entire population before 100 days because the inhibitory agent for diapause development appears to be more rapidly removed at lower temperatures [25,33,34]. Thus, delayed hatching at −6 and −12 °C, compared with at 0 and 6 °C, may have been due to the temperature being too low to initiate post-diapause development after the end of diapause, rather than the requirement for chilling not having been adequately satisfied. Many individuals began post-diapause development at 12 °C, as seen in the development completion model, but some eggs probably did not experience sufficient chilling when considering the lower survival relative to 6 °C and delayed hatching in terms of development time.

Lee and Lee [5] reported that adult density monitored by pheromone traps is a good predictor in estimating population density in the following year because the number of egg masses is highly correlated with the trap catches of adult males. This indicates that the outbreak potential could be identified in advance with a field survey of the previous year. In this respect, the assessment of the suitability of the winter chilling temperature, combined with adult density, could provide support for the forecasting of the outbreak possibility in the following season. Additionally, it is necessary to evaluate the continuity of preferable winter conditions in the long term because the population dynamics of *L. dispar* generally show a sequential process of (i) innocuous, (ii) release, (iii) outbreak, and (iv) decline [4,5]. Even if the population is in a release phase, unfavorable winter conditions would delay the outbreak. The variation in development time according to the winter temperature may affect the population process of *L. d. asiatica* in a more complex manner than the response in terms of the survival of overwintered eggs. Egg hatching must occur less rapidly to survive low temperatures, but early enough to synchronize with the bud burst of host plants and to complete the phenological process. The timing of egg hatching thus affects the successful emergence of adults and oviposition, consequently determining the population size in the next year. Therefore, the cold resistance of young larvae or the seasonality of preferred hosts needs to be considered in the assessment of the phenological barrier caused by the thermal response. This information would be helpful in estimating the potential invasive range as well as outbreak potential.

## 5. Conclusions

In summary, this study examined the effect of chilling temperatures on the survival and post-diapause development of *L. d. asiatica* eggs, collected from the field in November, taking into consideration the duration of winter temperatures in Korea. In the given temperature range, exposure for 100 days significantly affected both the survival and development rates of overwintering eggs. The survival rates declined at an extreme temperature of $-12\ ^\circ$C, and the development rates were accelerated as the chilling temperature increased. This information could provide support in assessing the outbreak potential in native regions and the possibility of range expansion in non-native regions, although other biotic and abiotic factors would also affect the mass occurrence and establishment [35]. The combination of the accurate assessment of egg hatching with the phenological parameters of larvae development would help in establishing: (1) management plans for susceptible second-instar larvae; (2) an effective surveillance program for adults in ports, for quarantine purposes; and (3) the potential range of expansion in non-native regions for the Korean population of *L. d. asiatica* in recommending against future introduction.

**Author Contributions:** Conceptualization, C.Y.L., J.-K.J. and Y.N.; methodology, C.Y.L., J.-K.J. and Y.N.; software, M.-J.K.; validation, M.-J.K. and K.E.K.; formal analysis, M.-J.K. and K.E.K.; investigation, C.Y.L., J.-K.J. and Y.N.; resources, J.-K.J., Y.P. and Y.N.; data curation, M.-J.K., Y.P. and Y.N.; writing—original draft preparation, M.-J.K.; writing—review and editing, M.-J.K., K.E.K., J.-K.J. and Y.N.; visualization, M.-J.K.; supervision, J.-K.J. and Y.N.; project administration, Y.N. All authors have read and agreed to the published version of the manuscript.

**Funding:** This study was supported by the National Institute of Forest Science (Project No.: FE0703-2022-01), Republic of Korea.

**Data Availability Statement:** The data presented in this study are available on request from the corresponding author. The data are not publicly available due to institutional policy.

**Conflicts of Interest:** The authors declare no conflict of interest.

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
