# Peer review of "Effect of Chilling Temperature on Survival and Post-Diapause Development of Korean Population of Lymantria dispar asiatica (Lepidoptera: Erebidae) Eggs"

_forests, doi:10.3390/f13122117_

Round 1

Reviewer 1 Report

This paper adds some important information on the Korean spongy moth populations. The methods are sound, although collecting in November does create some problems that the authors need to deal with a little more than they have already. There are some errors (citations, etc.) and omissions in the methods in the paper that indicate the authors need to pay attention to details a little more. More specific comments follow.

General Comments: Instances of poor English grammar and sentence structure throughout.

Title:  should include “Korean” before the species name since this study is specific to just that region and does not address all Asian areas where the insect occurs.

Introduction:

Line 38 The authors should make clear what invaded areas they are discussing.

Line 48 give refence for non-diapause hatching

Line 64 Author name miss spelled here and elsewhere.  Should be Keena not Kenna.

Methods:

Please provide a graph that shows the locations of the populations sampled. It would also be helpful if the map had a gradient that shows the lowest temperatures experienced across the landscape.

Line 87 Eggs collected in November likely also experienced some chilling since anything below 15 C can be considered as chill.  Since you collected from different regions the different populations may have received different amounts of chill. You can pull weather records and provide some information on these differences since they could affect the results.

Line 123 This is unfortunate. It would have been better to check for hatch at 12 C and get the daily hatch rate and start time. What proportion of the eggs had already hatched before the 100 days was completed?

Figure 1 what location is this for? If it is the average for the country, then it is not that informative. Are the minimum and maximum observed temperature?  What months were included in the winter?

Data Analysis: More details are needed on the methods used. What software program was used? What model was used? Was the development time data normally distributed? Frequency data can be evaluated using a Beta distribution and GLIMMIX in SAS followed by a Tukey difference test rather than the Chi-square method.  This is a more rigorous evaluation.

Table 2 Does developmental time include the 100 days in the temperature treatments?  If so it should not and should only be the number of days at 25 C needed for hatch.  Does the survival rate for the 12C include the hatch that occurred at 12 C? For consistency it should be included since those larvae did survive and hatch.

Table 2 and Figure 3 duplicate some data.  A graph with letters for statistical significance would be preferred and then just include the rest of the data in a table.

It would be very informative to know how much between egg mass variation there was in both % hatch and days to hatch at 25 C. A simple calculation of means of days to hatch for each egg mass and then providing data on that distribution would expose any substantial variation that was present.   I would expect the most variation at the extreme temperatures.

Discussion:

It would be helpful to have some discussion of how the results at temperatures like ones previously investigated compare. I believe that there is prior data for 5, 10 and 15C for other Asian spongy moth populations. Also, I believe that there is some information on how a Korean population compares to the far east Russian one the authors have referred to already. This might help the reader place the data in the larger framework of all the Asian populations.

Talking about the differences in the low temperatures experienced at different sites where eggs were collected compared to the percentage hatch at the sites would be good to add. I would expect that colder sites would have individuals that tolerate colder temperatures. This however could have been affected by longer chill times before collection that would have put the eggs in a different developmental stage when they were exposed to the treatments. Some discussion of the effects of the different temperature exposure before collection should be added beyond what is in the first paragraph.

Reviewer 2 Report

The current MS by Min-Jung Kim et al., entitled in “Effect of chilling temperature on survival and postdiapause development of Lymantria dispar asiatica (Lepidoptera: Erebidae) eggs” provides detail data to test the effect of chilling temperature (-12, -6, 0, 6, and 12 °C exposed for 100 days, and then incubation at 25°C) during the diapause phase on the survival and postdiapause development of L. d. asiatica eggs collected before winter to characterize their thermal response. And the results shown that exposure to chilling temperatures significantly affected both the survival and development times of overwintering eggs in the given temperatures. The survival rates declined at -12 °C, and the development rates accelerated as the chilling temperature increased. 

The topic of the manuscript is interesting and may helpful for the population estimation after post-diapause. The experimental design is sound, yet the analyses are not detailed enough and thus a little bit quite confusing. As, one of the main hypotheses (e.g., the effect of treatment temperatures on survival) was only directly tested the ultimate rate in chi-square test and Bonferroni. How about survival dynamics, such as using survival analysis in Log-rank (Mantel-Cox)? The hypotheses are not clearly defined, For instance, how about the optimal treatment temperature in the protocol, these results are also not discussed (e.g., significant effect of temperature in the protocol). And, for the two-parameter Weibull cumulative function model, the meaning about estimated parameters of a and b was not well defined and explain. In its current form, I recommend that a major revision is warranted.

Minor comments:
1. The protocol schematic diagram of temporal pattern in Figure 2 only show 0 °C, I suggest that all the treatments should be added in. 

2. The effect of low temperature treatment on the synchronous should be discussed. And, Line 200, Lin 228. Based on the current results, it can be seen that the threshold temperature for post-diapause development should below than 12 °C, above than 6 °C, as it is shown in Figure4. 

3. It can be seen that all the sample site in the current work in Korea are the suitable potential range for overwintering survival, however, for the estimating of the potential invasive range of the current pest, the current data cannot provide enough strength.

4. Line 188-189: At 12 °C, a substantial number of eggs ranging from 30.3 to 94.9% of total successful hatches in the treatment hatched before being moved to 25 °C, the percentage show here is not accordingly to the data of Hatches before incubation at 25 °C (%) in table 2, am I understood right?

Round 2

Reviewer 2 Report

I have carefully reviewed the revised paper forests-2040836 and their revision notes to the comments specified by the reviewers. 

Authors have suitably revised their MS according to referee' suggestion, it seems to be improved evidently.